# Investigating the Mental Health Impacts of Climate Change in Youth: Design and Implementation of the International Changing Worlds Study

Ans Vercammen [1,2,*], Sandhya Kanaka Yatirajula [3], Mercian Daniel [3], Sandeep Maharaj [4], Michael H. Campbell [5], Natalie Greaves [5], Renzo Guinto [6], John Jamir Benzon Aruta [6,7], Criselle Angeline Peñamante [6], Britt Wray [8] and Emma L. Lawrance [9,10,11]

1    The School of Communication and Arts, The University of Queensland, St. Lucia, Brisbane, QLD 4072, Australia
2    The Centre for Environmental Policy, Imperial College London, London SW7 1NE, UK
3    The George Institute for Global Health, New Delhi 110025, India
4    Faculty of Medical Sciences, The University of the West Indies, St. Augustine 685509, Trinidad and Tobago
5    Faculty of Medical Sciences, The University of the West Indies Cave Hill Campus, Bridgetown BB11000, Barbados
6    Planetary and Global Health Program, St. Luke's Medical Center, College of Medicine-William H. Quasha Memorial, Quezon City 1112, Metro Manila, Philippines; penamante.cc.e@slmc-cm.edu.ph (C.A.P.)
7    Counseling and Educational Psychology Department, De La Salle University, Malate, Manila 1004, Metro Manila, Philippines
8    Department of Psychiatry and Behavioral Sciences, Stanford Medicine, Palo Alto, CA 94305, USA; bwray@stanford.edu
9    Institute of Global Health Innovation, Imperial College London, London SW7 2AZ, UK
10   Grantham Institute for Climate Change and the Environment, Imperial College London, London SW7 2BU, UK
11   Mental Health Innovations, London W10 9FE, UK
*    Correspondence: a.vercammen@uq.edu.au

**Abstract:** As climate change continues unabated, research is increasingly focused on capturing and quantifying the lesser-known psychological responses and mental health implications of this humanitarian and environmental crisis. There has been a particular interest in the experiences of young people, who are more vulnerable for a range of reasons, including their developmental stage, the high rates of mental health conditions among this population, and their relative lack of agency to address climate threats. The different geographic and sociocultural settings in which people are coming of age afford certain opportunities and present distinct challenges and exposures to climate hazards. Understanding the diversity of lived experiences is vitally important for informing evidence-based, locally led psychosocial support and social and climate policies. In this Project Report we describe the design and implementation of the "Changing Worlds" study, focusing on our experiences and personal reflections as a transdisciplinary collaboration representing the UK, India, Trinidad and Tobago, Guyana, Barbados, the Philippines, and the USA. The project was conceived within the planetary health paradigm, aimed at characterizing and quantifying the impacts of human-mediated environmental systems changes on youth mental health and wellbeing. With input from local youth representatives, we designed and delivered a series of locally adapted surveys asking young people about their mental health and wellbeing, as well as their thoughts, emotions, and perceived agency in relation to the climate crisis and the global COVID-19 pandemic. This project report outlines the principles that guided the study design and describes the conceptual and practical hurdles we navigated as a distributed and interdisciplinary research collaboration working in different institutional, social, and research governance settings. Finally, we highlight lessons learned, specify our recommendations for other collaborative research projects in this space, and touch upon the next steps for our work. This project explicitly balances context sensitivity and the need for quantitative, globally comparable data on how youth are responding to and coping with environmental change, inspiring a new vision for a global community of practice on mental health in climate change.

**Keywords:** climate distress; eco-anxiety; climate crisis; interdisciplinarity; participatory research

## 1. Introduction

The literature detailing the (mental) health and wellbeing impacts of climate change is rapidly growing, as highlighted in a number of recent evidence reviews [1–9]. The proliferation of academic papers also reflects increasing interest from the practitioner and lived experience advocates who wish to understand the nature, prevalence, and outcomes of psychological responses to the climate crisis to better support clients and entire communities who are affected in various ways and to catalyse action [10–12]. A number of studies, including some large international surveys [13,14], have catalogued climate concerns, emotions, and the impacts on the lives and behaviours of people worldwide, with a particular interest in young people. These studies highlight that while some experiences are universal, there are contextual and cultural factors that shape or modulate responses to climate-related experiences. Ogunbode et al. [14], for instance, note that while climate change anxiety is negatively associated with wellbeing in almost all countries studied, there is more contextual variation in the links between climate anxiety, pro-environmental behaviour, and activism.

Capturing and understanding global differences and nuances in young people's psychological and behavioural responses to climate change is vitally important for informing appropriate mental health support and climate policies that work within the sociocultural context [3]. The idiosyncrasies of individual studies and the diversity of metrics to capture climate-related subjective experiences raise challenges for comparison and standardisation and make it difficult to draw conclusions about what substantive differences in the local climate psychology are and what is linked to incidental differences in assumptions, methodology, or specifics of the metrics employed. There is a clear need for collaborative investigations in which researchers working in different parts of the world, in partnership with their local communities, can share knowledge and methodologies and create contextually modified but conceptually uniform studies that allow us to compare and contrast insights derived.

This paper reports on the process of developing a coalition of researchers working on climate psychology and mental health in different sociocultural settings, namely, the UK, USA, Philippines, India, and three Caribbean countries. We detail how each group worked in partnership with local young people to amend and implement a survey with which to understand the thoughts and feelings of young people in response to climate change in the context of other (locally relevant) health and social crises (particularly the COVID-19 pandemic). Our aim is to share the principles we used to govern the research, the conceptual and practical challenges we have faced so far, and the observed strengths and limitations of our approach. From this, we derive some interim recommendations to set a firm direction towards creating an effective global community of practice in climate change and mental health research.

## 2. Climate Cares and the "Changing Worlds" Approach

The Changing Worlds study was initiated by Climate Cares, a collaboration between the Institute of Global Health Innovation and the Grantham Institute, both based at Imperial College London (https://www.imperial.ac.uk/global-health-innovation/what-we-do/research/mental-health/climate-cares (accessed on 20 March 2023)). The initiative brings together an interdisciplinary team of researchers, designers, policy-makers, and educators aiming to understand and support mental health during the current climate and ecological crises, supported by a globally representative advisory board of experts in psychology and psychotherapy, climate and health policy, planetary health, environmental epidemiology, public health, and science communication, among the wide range of relevant disciplines. Climate Cares' vision is to equip individuals, communities, and healthcare systems with

the knowledge, tools, and resources to be resilient to the mental health impacts of climate change and enable action that simultaneously benefits human health and wellbeing and the planet. Climate Cares does this through a programme of research, education and awareness raising, policy advocacy, and intervention design.

The original "Changing Worlds" survey was the first empirical research study conducted by Climate Cares and was developed in the UK in 2020, when there were no published data on the nature, prevalence, and severity of the mental health impacts of climate awareness in young people, the range of psychological responses they were experiencing, and the interactions with agency and behaviours. The study team included experts in environmental psychology, public health, mental health, neuroscience, climate and environmental science, and science communication. Additional input was sought from mental health practitioners and clinicians, and, crucially, from a young persons' advisory group (YPAG) specifically recruited for this study. In what follows, we briefly outline the development of the Changing Worlds study in the UK (Section 2.1) and describe the development of the research partnerships and international implementation of the study (Sections 2.2 and 2.3) and the dissemination of the findings (Section 3). Finally, we reflect on the challenges we faced and resolved and highlight prospects for the path forward in this field of research (Section 4).

### 2.1. Conceptualisation and Operationalisation of the Changing Worlds Study UK

#### 2.1.1. Rationale

The aim of the UK Changing Worlds study was to assess young people's psychological responses relating to the perception of climate change, primarily using quantitative metrics. We also included open-ended questions to potentially gather new insights into how climate change awareness influenced young people's thoughts, feelings, behaviours, and hopes and plans for the future in ways that were not captured by existing scales and questionnaires. At the time, there were no available data on the nature, prevalence, and severity of psychological responses to the climate crisis, though young people appeared—at least anecdotally—to be a particularly vulnerable group to mental health challenges and distress arising from climate crisis awareness. As the COVID-19 pandemic emerged as a major global health crisis at the time of the study, the decision was made to compare and contrast how young people in our UK sample responded to the two issues. Notably, the study was conducted during the first year of the pandemic, when public health interventions (e.g., lockdowns and requirements to isolate, wear masks, social distancing) in the UK were in full force. We screened for mental health and wellbeing indicators and collected basic demographic data. For each identified issue (i.e., the climate crisis and the COVID-19 pandemic), we asked about specific positive and negative impacts, as well as an assessment of the overall perceived impact severity. We measured psychological distress and emotions linked to the specific issue, their mental health status, and the self-reported extent to which this affected respondents' daily functioning and subjective wellbeing. Finally, we asked young people about their sense of agency in addressing each issue and about their actual engagement in pro-environmental and climate action. Established concepts were measured using previously published scales available at the time (e.g., "climate distress" was measured with Reser et al.'s [15] scale). Scales developed for one issue were adapted with minimal rephrasing to be comparable for the other issue, and this was achieved with the support of the YPAG (see below for further detail). Complementing the quantitative survey elements, a small number of open-ended questions were included. This was intended to provide richer insights into the subjective experience of climate distress, respondents' motivations for acting on it, and reflections on the impact of climate change in their lives, including their hopes and fears for the future. It was anticipated that the free-text responses would be thematically analysed to detect novel patterns or conceptual directions which would aid in setting and prioritising future research questions.

### 2.1.2. Conducting Research with Young People

While children, adolescents, and young adults are all considered to be particularly vulnerable groups, for the Changing Worlds study, we chose to focus on the older adolescent/young adult group (16–24 years) for conceptual and practical reasons. The first reason is that the adolescent and early adult years are also key periods of vulnerability for the development of mental health conditions [16]. The dual challenge of a global pandemic and the ongoing climate crisis was presumed to be a particularly significant factor for people at this sensitive developmental stage. More practically, working with young children requires additional ethical and logistical considerations, and gaining access to participants would have required collaborating directly with schools or other community groups. Under pandemic restrictions, this would have been extremely difficult. Adolescents and young adults, on the other hand, are highly technology literate and have a strong online presence, making them an easily accessible group who would also be able to independently decide whether or not to take part in the study, without the need to engage a gatekeeper. For the UK study, this meant we could include young people aged 16 and over. As common law in the UK presumes minors between 16 and 18 years of age are usually competent to give consent to medical treatment, they are also generally assumed to be competent to consent to participation in research without the need for parental or guardian consent. Further design considerations were directed by local ethical guidelines (e.g., those set out by the UK's Economic and Social Research Council) which highlight that "researchers should ensure that risk and harm in research is minimised and that adequate protection of children and young people is ensured. They should also consider the ethics implications of silencing and excluding children from research about their views, experiences, and participation" (https://www.ukri.org/councils/esrc/guidance-for-applicants/research-ethics-guidance/research-with-children-and-young-people/ (accessed on 20 March 2023)). Climate Cares supports the principle that young people, who are disproportionately burdened by the climate crisis, should be afforded a voice in the matter. This includes providing opportunities to exercise agency with respect to research that is relevant to them, while also accounting for young people's capabilities and vulnerabilities, including their mental and physical health. We provided participant information materials in accessible language to ensure that those interested in taking part understood the procedures and the consequences of their participation, and that they were able to give prior informed consent without the need for additional parental/guardian consent. We also provided a broad range of freely accessible resources, online information, and helplines whereby young people can access support for mental health concerns. The study proposal was approved by the relevant research ethics committee at Imperial College London, which evaluated the risk level as appropriate for the study population, given the safeguards that were in place.

We recognised that when it comes to mental health and wellbeing, young people are experts in their own experiences and that it was vitally important to not just collect information from them but to engage young people directly in the design of the study. The YPAG, which consisted of a diverse group of ten young UK residents of different ethnicities and ages living in different parts of the country, provided feedback on the content and presentation (e.g., length, wording) of the survey and was also consulted on the design of the questions. For instance, the YPAG helped to create a list of the perceived positive and negative impacts of climate change based on a previously published questionnaire about the possible positive and negative impacts of the COVID-19 pandemic on the lives of young people. This was to ensure that the survey captured the impacts most relevant to the target sample and addressed their needs and concerns. The YPAG consultations were held over a series of online (Zoom) meetings, and participants were reimbursed for their time in line with UK Standards for Public Involvement in Research.

### 2.1.3. Sampling and Distribution

In the UK, our initial approach was to reach out to youth via targeted participation calls on websites, existing mailing lists, and social media linked to youth groups (e.g.,

religious, community groups), Universities, schools, mental health charities, and other organisations within the researchers' professional networks, as well as some snowball sampling. These participants were entered into a draw to win high street gift cards. Small rewards or raffles such as these are considered appropriate compensation without introducing undue inducement. We were deliberate in approaching a broad range of youth-focused groups to distribute information about the survey, so that not only young people already engaged in climate-related activities would be recruited. While our recruitment drive was moderately successful, to reach our target sample, we additionally distributed the survey using a paid survey panel service (Prolific). We set no participation restrictions other than the age, residency criteria, and self-assessed English language proficiency. These participants received immediate monetary compensation using the remuneration standard set by the company. Online research panel services have become well-used recruitment tools, especially in social research. They are now widely available, and many give researchers access to a broad demographic range of participants in countries such as the UK, the USA, Canada, and various European nations. There are limitations to their use as the quality of respondents has been called into question, and the majority of panellists do reside in a minority of countries. Nevertheless, for the UK study, this approach proved fruitful compared to the community sample, which saw a relatively high drop-out rate in the early stages of the survey (e.g., people clicked on the survey link, but did not complete the questions), which is common in web-based research [17].

*2.2. The "Changing Worlds" Research Partnerships*

2.2.1. Principles of Collaboration and Interdisciplinarity

We established partnerships between the UK Climate Cares team and several researchers/research groups involved in Planetary Health, located in the USA, the Caribbean (Trinidad and Tobago, Guyana, and Barbados), India, and the Philippines. These partnerships were initially facilitated through the professional networks of the Climate Cares team and largely based on verbal rather than formalised agreements. The guiding principle was a common interest in building a richer global understanding of young people's responses to the climate crisis. The partner organisations each agreed to develop a localised version of the Changing Worlds study according to their interest and needs, in a way that preserved reasonable equivalence across settings. Each lead researcher recruited a local network of researchers and practitioners with relevant skills and knowledge, recognising the need for interdisciplinarity as this type of research crosses boundaries between public health, climate change studies, environmental psychology, psychiatry, epidemiology, human geography, and possibly other disciplines. They also recruited local Young Persons' Advisory Groups to ensure materials and methods met local understandings and needs. The composition of the research teams at each site, therefore, varied, depending on available local resources and capacity. The UK team formed the central connection, facilitating interaction among the various site teams, which then allowed for greater access to the broad expertise required to successfully implement, interpret, and widely disseminate the research outputs and outcomes.

2.2.2. Funding and Project Management

Each of the site teams secured independent funding or allocated internal funding for the implementation of the project at that specific location. This decentralised funding approach was a necessity as we did not have access to sustained and sufficient external research funds for a multi-site investigation. As the mental health impacts of the climate crisis is an emerging field of research, there has been very limited relevant external grant opportunities to date, challenging the resourcing and pace of such research. As each site team managed their resources independently, the different studies operated on their own timelines. The resulting staggered implementation had certain benefits as it allowed the site teams to exchange lessons learned at varying project stages so that issues or challenges could be avoided or resolved more quickly. For instance, documentation prepared for,

and approved by local Internal Review Boards was shared with other teams to facilitate ethical approvals.

In line with the independent funding arrangement, each site team was responsible for the local project management. Throughout the project cycle, the UK team provided a supporting role and offered additional capacity and human resources where requested, e.g., to aid with data analyses and manuscript preparation and to assist in study design and set up, as well as providing materials such as YPAG recruitment resources that could be locally amended. The extent of this support varied depending on the needs and resourcing of the site teams. For instance, the UK team was an equal partner in the USA study and a consulting/advisory partner in the India study. The site teams provided in-depth contextual and operational knowledge to successfully design and distribute the surveys and interpret the findings with reference to the relevant international and regional literature. We held regular web-based meetings between the UK team and the individual site teams, as well as (less frequent) multi-team meetings to share project updates.

### 2.3. Local Adaptations of the 'Changing Worlds' Methodology

Table 1 provides a summary of the study characteristics across the five different settings, highlighting survey design decisions, stakeholder involvement, recruitment strategies, sample characteristics, operational factors, outputs, and/or dissemination plans for future outputs. Each study received local ethical approval (at the time of writing, the Philippines study team was in the process of obtaining ethical approval from their IRB, and data collection had not yet started); at all sites, respondents gave prior informed consent; only participants able to give consent themselves were recruited for the studies. The age range varied by site, as local consent regulations with regard to minors (<18 years of age) imposed different restrictions on recruitment.

### 2.3.1. Survey Design and Content

The UK Climate Cares team shared the original survey instrument with the site teams. The survey was then tailored through discussions with locally recruited YPAGs. At most sites, the YPAG participants were selected from a larger pool of respondents to an open call for Expressions of Interest, or through more targeted advertisement of the opportunity to relevant youth organisations such as student groups and youth climate or social justice advocacy groups. Although we did not consistently apply formal selection criteria, we generally aimed to achieve reasonably gender balance within each YPAG and sought to include participants from different geographic regions (e.g., urban/rural), cultural/ethnic backgrounds, and socioeconomic groups, as relevant to the country. We also asked prospective YPAG members to indicate their availability and motivations for participation to identify individuals that were sufficiently engaged to ensure a productive YPAG process. The India site team employed an existing adolescent expert advisory group that had been set up for a separate study. Additional changes to the survey were based on local researchers' contextual knowledge and relevant academic and grey literature sources. These local adaptations largely fell into three categories, namely: (1) amendments to the emphasis on crises in addition to climate, for example, whether to include questions about the COVID-19 pandemic or other locally pertinent challenges (e.g., post-election political developments in the Philippines); (2) alterations (including additions or deletions) to ensure cultural relevance, such as what emotions made sense to the local young people; (3) wording or language changes to be equivalent but presented in a different language/culture. The first category included the addition of questions addressing pertinent social issues that intersect with climate change and public health crises, while the second included the omission of items considered inappropriate or likely to be interpreted in a manner inconsistent with the intended purpose of the original survey. For instance, we were interested in the kinds of emotions young people associated with climate change (and the COVID-19 pandemic), but after review by the experts and YPAG from the George Institute India, it was decided to omit certain emotion words from the list in the survey

implemented there. These included, for instance, guilt and disgust, as these were words not typically associated with climate change or else they lacked a local translation that made sense in the context of climate change responses for the young slum dwellers targeted in the Indian version of the study. The USA YPAG, on the other hand, advocated for the inclusion of additional climate-related emotion words, including "cynical", "numb", and "manipulated", which reflected their local experiences. In the Caribbean Region, the local YPAG conducted a pilot amongst peers to test the readability and contextual relevance of survey items, and only minimal changes were suggested. Where relevant, the survey was translated into local languages to ensure that the participants' comprehension of the questions was not hindered by language barriers, and back translations were conducted to ensure linguistic equivalence (for details, see Table 1).

### 2.3.2. Sampling, Recruitment and Data Collection

The sampling and recruitment methodology was tailored to tap into locally available resources. Because regulatory and normative considerations were not uniform across countries, we adjusted procedures as needed. For example, in most sites, we limited participation to research volunteers between the ages of 18 and 24 as the regulatory frameworks for research with minors differed from the UK, making the inclusion of 16–18 year-olds impractical. Due to the success of the panel service approach in the UK, we used a similar service provided by the survey company Qualtrics in the USA where we had the funding to do so. This allowed us to set specific selection criteria and to create a sample that mirrored the ethnicity, age, and gender breakdown according to the USA census. We were also able to over-sample from geographic regions that had recently experienced climate-related extreme weather events or were subject to other (e.g., chronic) climate change phenomena such as drought or air pollution. We identified such regions at the county level as reported in the Federal Emergency Management Agency (FEMA) National Risk Index for Natural Hazards, the National Lung Association State of the Air Report, and the Centers for Disease Control (CDC)/Agency for Toxic Substances and Disease Register (ATSDR) Social Vulnerability Index. These additional restrictions significantly increased the cost of recruitment but also substantially reduced the data collection time. At other sites, e.g., in India, the research team had access to a database linked to a previous study comprising 9905 households, who agreed to be contacted for further research. The project sites were purposely and pragmatically selected because they were urban slums located close to the research team's offices in the cities of Faridabad and Hyderabad. This allowed the team to easily recruit from a unique population. An internet-based survey was not feasible; instead, trained field investigators collected the survey data in person. In the Caribbean countries and the Philippines, social media is very popular among the target demographic and mobile phone access is very common. This facilitated the recruitment of participants through social media marketing, including via the use of relevant local influencers known to many young people and data collection through a mobile-friendly online survey.

**Table 1.** Summary of the Changing Worlds study characteristics.

| | UK | India | Caribbean | The Philippines | USA |
|---|---|---|---|---|---|
| Lead organisation | Imperial College London | The George Institute for Global Health India | The University of the West Indies | St. Luke's Planetary and Global Health Program | Stanford University Center for Innovation in Global Health |
| Research team composition | The UK lead is based at the Institute for Global Health Innovation. Within-institution collaboration was established with the Grantham Institute for Climate Change and the Environment, and additional input was provided by researchers from the Centre for Environmental Policy, and the School of Public Health. | The George Institute for Global Health India was the lead organisation for the study | The Caribbean lead organisation was The University of the West Indies, St. Augustine, and Cave Hill Campuses. Partners were also brought in from the University of Guyana and the College of Caribbean Family Physicians to support execution of the study. The team was assisted by a Queen Elizabeth Fellow from McGill University and supported by the UK study team. | The Philippines team comprises a globally recognised expert in Planetary Health (MD), a counselling practitioner, and an environmental psychology researcher, supported by junior researchers with multi-disciplinary training. The team was in the process of recruiting additional members to support the rollout of the survey and subsequent analysis. | The US lead recruited local data analysis support and epidemiology/biostatistics expertise from within the leading institution. Additional input was sought from a US-based climate-aware psychiatrist with a deep interest in young people's mental health and climate agency to support the conceptual stages of the study and report writing. The UK team was an equal partner in the study. |
| Disciplinary representation | Psychology; neuroscience; climate science; design; science communication; mental health; psychiatry; epidemiology | Psychiatry; public health; mental health | Medicine; public health; psychology; pharmacy; planetary health biostatistics | Planetary health; public health; medicine; psychology; environmental science; epidemiology; biostatistics | Psychology; mental health; planetary health; science communication; epidemiology |
| Study dates/status | 2020–2021 Complete | 2021 Complete | 2021–2022 Complete (The survey was rolled out across Guyana, Trinidad and Tobago, and Barbados with IRB approval from the University of the West Indies St. Augustine and Cave Hill, the University of Guyana, and the Pan-American Health Organization (PAHO). PAHO has been spearheading a separate ethical approval process for a separate roll-out of the survey in Jamaica, which is anticipated in the latter part of 2023.) | 2023 | 2021–2022 Complete |
| IRB approval | Imperial College Research Ethics Committee—approval number 20IC6060 | Independent Ethics Committee of The George Institute for Global Health New Delhi, India—Ref. no. 03/2021 | Trinidad and Tobago (CREC-SA.0941/05/2021, PAHOERC.0375.02); Guyana (IRB #107/2021); Barbados (IRB #210807-B) | As of the time of writing, being reviewed by the Institutional Ethics Review Committee of the St. Luke's Medical Center, pending approval | Stanford University Institutional Review Board (IRB)—eProtocol #62589 |
| Co-creation and stakeholder involvement in study design | Survey was drafted by Climate Cares researchers and reviewed/edited by ■ Young Persons Advisory group, consisting of 10 young people from diverse social, ethnic, and cultural groups in the UK; ■ Expert advisory group; ■ Clinical advisor (youth mental health specialist). The YPAG was consulted on the content (e.g., list of relevant climate and COVID-19 impacts, the emotions associated with both issues), formatting, wording, and length of the survey, | Pre-existing adolescent expert advisory group (AEAG) that was part of another research study entitled "Adolescents' Resilience and Treatment Needs for Mental Health in Indian Slums (ARTEMIS)" | Ambassadors were recruited from students at The University of the West Indies Cave Hill Campus (Barbados), St. Augustine Campus (Trinidad and Tobago), and The University of Guyana, or students in the Environmental or Medical Science Programmes at The University of the West Indies and The University of Trinidad and Tobago, or young people involved in climate change NGO's in the Caribbean; The original survey was piloted with small sample (N = 30) and feedback was used to make changes to/contextualise the survey content. | The young person's advisory group (YPAG) was recruited through nominations and referrals by major youth advocacy networks and organisations, consisting of eight young people representing different advocacy networks (e.g., mental health, climate change, social justice), regions of the country, and specific populations, including Indigenous representation. The YPAG was consulted on content, wording, translation into the Filipino language, formatting, and dissemination strategy. The YPAG also piloted the survey and gave feedback on length, questions, and factors that could contribute to attrition. | Young person's advisory group (YPAG), which consisted of nine racially/ethnically and geographically diverse young people from across the USA. They were recruited through youth advocacy networks and social media. The YPAG was consulted on content (e.g., list of emotions), wording, and formatting. The YPAG piloted the survey and gave feedback on length, questions, and factors that could contribute to attrition. |
| Survey topics | Climate change; COVID-19. | Climate change; COVID-19. | Climate change; COVID-19. | Climate change; COVID-19; Politics and election stress. | Primary focus on climate change; COVID-19. |

**Table 1.** *Cont.*

| | UK | India | Caribbean | The Philippines | USA |
|---|---|---|---|---|---|
| Survey content and local variations | ■ **Demographics:** age, gender, LGBTQ+ identification, ethnicity/cultural background, socioeconomic status (Family Affluence Scale), location (urban/rural), living arrangement;<br>■ **Symptoms of mental ill health:** stress (PSS), anxiety (GAD-7), depression (PHQ-9);<br>■ **History of mental ill health/diagnoses:** specific diagnosis/diagnoses if applicable;<br>■ **Overall life satisfaction:** single item rating scale 0–10;<br>■ **Personal circumstances relating to COVID-19:** current pandemic measures; time spent outside; COVID-19 infection status (self/loved ones); pandemic-related interruptions to work, social and personal life;<br>■ **Positive and negative impacts:** Separate, pre-defined lists of direct physical and psychosocial impacts relating to climate change/COVID-19;<br>■ **Psychological responses:** 18 different emotions, Distress Scale, Agency Scale, interference on wellbeing in relation to climate change/COVID-19;<br>■ **Action-taking:** engagement in (climate) activism before and during pandemic, participation in pro-environmental behaviours before and during pandemic;<br>■ **Hopes/concerns for the future:** relative worry about factors that will affect personal future prospects (e.g., politics, economy, climate change), list of priorities for post-pandemic recovery;<br>■ **Open-ended questions:** personal experiences with/impacts of COVID-19, personal experiences with/impacts of climate change, participation in climate activism, personal actions taken against climate change, hopes/fears for the future. | ■ Demographic questions were adapted to the Indian context; socioeconomic status was assessed using the Kuppuswamy's socioeconomic status scale [18]<br>■ Questions about mental health history omitted due to sensitivity around this topic;<br>■ Some climate impact questions were removed as irrelevant in India, e.g., positive impact of "enjoying more sunshine and warmer weather";<br>■ Collapsed and reworded response categories from 5-point Likert scale to 4-point Likert scale to facilitate interpretation;<br>■ The emotions "interested", "disgusted", "outraged", "guilty", "frustrated", "concerned", "disconnected", "apathetic", and "engaged" were not included in the emotion rating questions as they were not associated with climate change and/or did not have a direct translation that made sense in the context of climate change responses;<br>■ No open-ended questions due to concern over data quality based on previous experience with collecting open-ended interview data in similar sample. | ■ Given specific demographics of the Caribbean, ethnicity/cultural background included Afro-Caribbean, Indian-Caribbean and other mixed-ethnicity groups;<br>■ The Family Affluence Scale was modified to the Caribbean context, with modifications of proxy markers of affluence. | ■ Most demographic questions from the UK study were adopted with addition of questions on disability, membership in Indigenous communities, and participation in environment-related sectors (e.g., farming);<br>■ Questions regarding economic status were revised to suit the Philippine context;<br>■ Other sections from the UK study (e.g., wellbeing assessment, history of mental ill health, involvement in climate action, hopes and concerns for the future, etc.), including mental health screening (PSS, GAD-7, PHQ-9) were retained;<br>■ Similar to the US study, Hogg Eco-Anxiety scale and questions about psychological adaptation were added;<br>■ COVID-19-related questions were minimised because the survey will be administered after the lifting of the public health emergency, but questions related to psychological responses to recent political developments in the country were included;<br>■ A short section inquiring about potential coping activities that are culturally relevant to contemporary Philippine context—such as praying, social media use, and karaoke singing—is also included;<br>■ Open-ended questions were mostly removed to focus on generating quantifiable insights and to reduce the burden with responding to an already-lengthy questionnaire. | ■ Added several emotions to the emotion rating question based on input from the YPAG, including "cynical", "numb", "manipulated", "hate", and "exhausted", as this reflected their individual experience in the USA context, e.g., hate was felt towards climate deniers, cynicism was felt in relation to system change inertia;<br>■ Questions about participating in activism made no specific reference to any changes since the pandemic because the USA survey did not undertake a period of extensive lock-down or COVID-restrictions;<br>■ The agency question "I believe climate change is inevitable no matter what we try to do to stop it" was rephrased as "I feel the threat of climate change cannot be reduced, no matter what actions are taken now" based on YPAG feedback. The question was interpreted in different ways, i.e., as either referring to beliefs about climate change being a hoax, or that the influence of fossil fuel companies is so powerful that "we" (as activists) cannot make change. This interpretation seemed unique to the USA context where the fossil fuel lobby and denialism have historically been and continue to be powerful forces in the debate on climate change and climate action.<br>■ Addition of several scales and questions:<br>- Perception of having already being directly impacted by climate change (yes/no);<br>- Experience with specific extreme weather events/climate-linked events;<br>- Hogg Eco-anxiety Scale;<br>- Psychological Adaptation Scale;<br>- Meaning-focused Coping Scale;<br>- Effect of climate change on future plans, including family planning, financial decisions, where to live and travel;<br>- Trust in the political system. |
| Survey tool | Available on OSF: https://osf.io/9ewtn (accessed on 1 August 2023) | Available upon request | Available upon request | Available upon request | Available on OSF: https://osf.io/vr9xy (accessed on 3 July 2023) |
| Operational variations | ■ Implemented on Qualtrics | ■ Implemented on REDCap;<br>■ Translation into Hindi and Telugu, the local languages spoken in Faridabad and Hyderabad;<br>■ Surveys conducted face-to-face by trained field investigators;<br>■ Respondents were interviewed in their home. | ■ Implemented on REDCap;<br>■ Survey available in English only. | ■ Translation into Tagalog;<br>■ Survey also available in English;<br>■ Will be implemented on a Google Form. | ■ Implemented on Qualtrics;<br>■ Survey available in English only. |

**Table 1.** *Cont.*

| | UK | India | Caribbean | The Philippines | USA |
|---|---|---|---|---|---|
| Recruitment and sampling strategy | ■ Survey link was distributed through authors' professional networks and to various mental health and climate charities active in the UK;<br>■ Snowball sampling;<br>■ Paid research panel service (Prolific). | Participants were recontacted from a pre-existing census project database [19] | ■ Snowball sampling: ambassadors were asked to recruit from their social circles (meeting the eligibility criteria)—ambassadors also reached out to church groups, community groups and NGOs;<br>■ Social media marketing: online social influencers were asked to advertise the study to followers;<br>■ Publication of a paid online ad to gain further traction. | ■ Social media marketing via institutional social media accounts, as well as those of the networks represented by YPAG members and other networks connected to the research team;<br>■ Snowball sampling: YPAG members are encouraged to forward the survey link within their networks;<br>■ Respondents will also be asked to forward the questionnaire to other young people;<br>■ The survey link will also be sent as an email blast to all the contacts of the project team, including personal and professional networks, to reach other young people not included in the aforementioned social media accounts and networks. | Paid research panel service (Qualtrics) with sampling to mirror the ethnicity breakdown and gender parity according to the most recent census, as well as targeted oversampling from geographic regions subject to (recent) extreme weather events and air pollution. |
| Sample size and characteristics | ■ N = 530;<br>■ More women/non-binary (34% men);<br>■ Aged 16–24 (M = 21.0; SD = 2.53);<br>■ More ethnically/culturally diverse (71% white European) compared to general UK population;<br>■ UK-wide geographic distribution, but majority resident in England. | ■ N = 536;<br>■ 259 men/277 women<br>■ Aged 16–24;<br>■ Resident in urban slums in 2 regions (Faridabad in the North Indian state of Haryana and Hyderabad in the South Indian state of Telangana). | ■ N = 194 (Trinidad & Tobago); N = 86 (Barbados); N = 196 (Guyana); more women/non-binary (34% men);<br>■ Aged 18–24;<br>■ Ethnicity/cultural background mainly African Caribbean (39%); multiple ethnicities (29%); Indian Caribbean (27%). | ■ The target sample size is n = 385 based on a best estimate for the population size of young people ages 15–24 (20,295,493 as of 2019 as reported by the Philippine Statistics Authority) with a margin of error of 5% and confidence level of 95%;<br>■ Focus on the 18–24 age group. | ■ N = 2883;<br>■ Gender-balanced sample (43.8% men);<br>■ Aged 16–24 (M = 20.35; SD = 2.50);<br>■ Ethnically and culturally diverse;<br>■ Significant proportion identified as LGBTQI+ (26.8%);<br>■ Largely metropolitan area residents. |
| Academic outputs (as of August 2023) | 2 academic papers (1 published, 1 in press) | 1 academic paper published in Lancet- Regional Health Southeast Asia on 20 April 2023 | 1 academic paper (submitted to peer review journal) | - | 1 academic paper (submitted to peer review journal; preprint available) |
| Manuscript DOIs | https://doi.org/10.1016/s2542-5196(22)00172-3<br>https://doi.org/10.31219/osf.io/e3tpu (accessed on 1 August 2023) | https://doi.org/10.1016/j.lansea.2023.100191 (accessed on 1 August 2023) | - | - | https://doi.org/10.21203/rs.3.rs-2698675/v1 (accessed on 1 August 2023) |
| Data availability | https://osf.io/mgu6x (accessed on 3 July 2023) | Contact authors | Contact authors | Contact authors | Contact authors |
| Other dissemination channels | ■ Climate Cares social media accounts;<br>■ Talks attended by educators and policymakers such as the Department of Health and Social Care in the UK;<br>■ Participation in COP27. | The study findings were shared during a symposium held at the 9th World Congress of Asian Psychiatry (WCAP 2022) from 16–18 September 2022. | The National Health Research Conference of Trinidad and Tobago 2022. | - | ■ Gen Dread online community and newsletter (gendread.substack.com) and Gen Dread social media channels;<br>■ Lectures and workshops delivered to policymakers, researchers, clinicians, and activists. |
| Joint dissemination of findings | Network of Environmental Social Scientists seminar November 2022—https://www.nessaustralia.org/coping-with-eco-anxiety-in-a-pandemic/ (accessed on 1 August 2023)<br>PHA 2022 Side Event—https://youtu.be/tyW7YiE61uk (accessed on 1 August 2023) | | | | |

### 2.3.3. Data Analysis and Data-Sharing across International Borders

To facilitate data processing and assist with statistical analyses, the UK team made their data analysis plans, pre-publication drafts, and preprints of manuscripts available to the other site teams. In principle, Climate Cares will, as a collective, ensure that all publications resulting from its work are open-access to maximise the impact and ensure equitable access to the findings [20]. In a crisis discipline, such as the emerging nexus of climate change and mental health, ensuring that newly emerging evidence is freely accessible will benefit both scientific progress and the integration of evidence in policy and practice. Furthermore, we advocate for the adoption of a wider range of open science practices (e.g., preregistration of hypotheses, making research data and materials freely available, and publication on preprint servers). However, we recognise that there are different norms around research practices globally. For instance, while we anticipated that we would freely share the collected data among research teams to aid in data processing and interpretation, there were institutional policy barriers relating to the ethical implications of the international transfer of research data or even access to the data for researchers in other countries. Some of the information collected, while anonymised, is theoretically re-identifiable while a subject number database is held somewhere in the world (pseudonymised), thus requiring a high level of oversight and data security practices to ensure that the participants' rights and privacy are protected. Countries and institutions have different risk assessments around this process, and this will impact the transfer of personal data for research purposes. Recent developments in data protection regulations, such as the General Data Protection Regulation (GDPR; which governs the exchange of personal data for EU and UK researchers), require researchers to expend additional resources in order to confirm that other parties provide equivalent standards of protection for legitimate data transfers. This burden of responsibility, and the lack of training with respect to the various legal bases for data sharing, may make researchers hesitant to engage in data-sharing agreements in international research collaborations, especially when potentially sensitive data (e.g., health data) is collected [21]. We can only stress that prospective collaborators discuss their data sharing requirements early in the research partnership formation. Questions about who will own and control the data and how decisions will be made about potential future secondary data analyses should be included in these discussions. Most institutions will have the required expertise and capacity within their legal team and/or research office to facilitate these discussions and provide templates. Nevertheless, research agreements can take time to develop. It is therefore an important consideration in resource and timeline planning to avoid obstructions in the research process further down the line.

### 3. Dissemination of the Findings

Our decision to publish separate academic outputs for each of the sites was motivated by the staggered implementation of the different projects and our desire to share our observations in the most expedient manner. In line with this, as noted, where this was institutionally supported or even promoted, we published preprints of our manuscripts and provided open access to the survey tools and the data. Separate publications also allowed each site team to fully explore the nuance and detail captured in their survey results, including the qualitative analysis of free-text responses to provide rich contextualisation to the findings and to enhance the local policy relevance of the outputs.

To maximise our impact with the communities, institutions, and decision-makers that this research serves, Climate Cares and its various partners maintain an active profile on social media with which to disseminate key findings and project milestones such as publications. We also coordinate stakeholder engagement events and participate in scholarly meetings and discussions. Examples of engagement activities are provided in Table 1.

This initial approach—giving site teams precedence in terms of publishing their findings—does not rule out the possibility of a future publication reporting on the survey measures that were directly comparable, and exploring how differences in demographic

and sociocultural factors affect the outcome variables. For instance, one research question of interest is whether the association between climate distress and mental health indicators is identical across the different settings. Informal discussions among the teams suggest that this may not be the case, but this requires further analytical validation. Additionally, we are interested in comparing the relative burden of worry about climate change in relation to other everyday worries (such as work, finances, relationships, school), which we anticipate will vary depending on the economic, political, and social context, as well as the experience of direct (physical) climate impacts.

To provide an interim summary and comparison of the key findings, we organised a side event at the 2022 Planetary Health Alliance Meeting, which was attended either in-person or virtually by members of each of the site teams. Copies of the talks and slides are available from the authors upon request, and a video recording of the full session is publicly available (https://youtu.be/tyW7YiE61uk (accessed on 1 August 2023)). It is outside of the scope of this report to provide an in-depth comparison of the survey results or a quantitative analysis of the between-site differences. However, the high-level observations, summarised in Figure 1, serve to illustrate how our approach allowed us to uncover context-specific patterns within common thematic areas such as climate distress, emotions associated with climate change and the pandemic, and agency or action-taking. While this requires further validation, preliminary qualitative evaluation of the key findings suggests that the experience of climate distress is shaped by sociocultural and temporal variables (i.e., the timing of the survey). In the UK, for instance, while climate "distress" was relatively elevated, it had little effect on participants' day-to-day wellbeing compared to how their thoughts and feelings around the pandemic affected their everyday functioning. In India, young people appeared equally distressed about both issues and rated the effects on their day-to-day functioning as similar too. We hypothesise that this difference in perspective might be linked to the scarcity of physical climate impacts in the UK, and the timing of the survey in the midst of pandemic lockdowns in the UK. The distress scale we used measures general concern around climate change in a rather abstract way and can represent vicarious rather than direct personal experience. The distinction between this kind of abstract "distress" and more severe psychological impacts is supported by the findings from the US survey, where we additionally included an eco-anxiety measure. As in the UK, the overall US sample showed moderate-to-high climate distress, yet few reported symptoms such as rumination or affective disturbances that interfere with normal functioning. However, the US survey allowed us to explore how this might change when people are exposed to physical climate impacts. Young people reporting personal experience of events which they associated with climate change showed elevated scores on almost all outcome measures, including eco-anxiety. In India, both climate change and pandemic distress appeared to interfere with mental health, which may be because the specific sample (youth living in urban slums) may have less access to support systems, increasing the risk for poor outcomes in the context of any kind of public health crisis. We noted that anxiety and mental health concerns were common among Caribbean youth as well, and this was linked to perceptions of (limited) socioeconomic opportunity, a finding that seemed to be unique to that setting. At all sites, we noted differences in the types of emotions young people associated strongly with climate change versus the pandemic. Interestingly, positive emotions such as engagement and interest were common for climate change. While guilt was also noted at several study sites, pre-testing of the survey showed that this was not understood as a climate-related emotion in India. On the basis of our initial observations, further examination of cultural differences in the affective characterisation of climate change impacts seems warranted. Another important theme we explored was how feelings and thoughts around climate change might shape agency (the sense that one can effect change) and action taking (e.g., adopting pro-environmental behaviours). Distress about climate change (in the UK) and even experience of direct climate impacts (in the US) was linked with a greater sense of agency, suggesting that active hope can be maintained alongside, or even because of, engagement with the reality of climate

change. UK youth who were more distressed tended to report more pro-environmental behaviours (i.e., increased distress was linked to taking actions that conserve or protect natural resources or avoid GHG emissions). In India, despite perceptions of agency being high, actual engagement in climate action was low. To understand which factors act as facilitators and barriers on youth action-taking, more in-depth qualitative research may be needed.

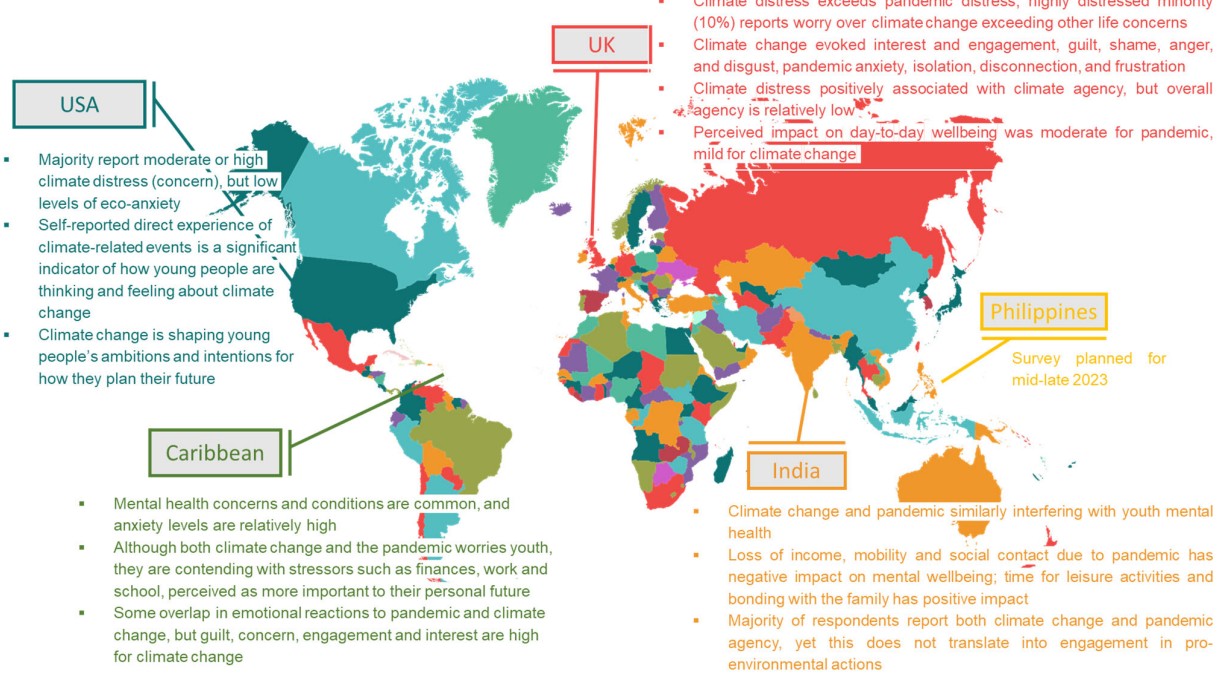

**Figure 1.** Summary of key findings from the completed "Changing Worlds" surveys in the UK, India, the Caribbean (Barbados, Guyana, and Trinidad and Tobago), and the USA. Data from the survey in the Philippines are not yet available.

## 4. Limitations and Recommendations

### 4.1. Working through Challenges as a Team

As is the case with any collaboration, but particularly in a decentralised partnership with multiple site teams, a priori and joint decision-making about the assignment of roles and responsibilities is crucial. For those working in a crisis discipline, the inclination to respond expeditiously to perceived research gaps and needs in the communities we serve may cause us to miss some of those pivotal planning steps. We did not always start with an explicit assessment of each site team's goals, available resources, and capacity, which complicated the setting of realistic timelines. With the additional challenge of working on limited budgets, this meant that the Changing Worlds study was at times under-resourced, resulting in delayed deliverables (e.g., academic papers) and evidence implementation. Even when pressed for time, we would advocate for conducting a SWOT (strengths, weaknesses, opportunities, and threats) analysis [22] to help prioritise resource-sharing and capacity-building activities across the teams. This does not have to be a formal or time-consuming process, but recording this in some form is advisable as this will promote accountability. We note the importance of considering this as a reciprocal process that recognises the unique skills, knowledge, and resources of each partner, rather than a unidirectional exercise where the lead organisation assumes a position of power and control over resources [23].

An additional challenge in international research collaborations involving north–south knowledge-sharing is the significant structural biases in academia which reflect global power relations rooted in colonial legacies. These entrenched patterns affect how we

produce and value knowledge, which, when driven by academics in the global North, often fails to fairly recognise the interests of the academics and stakeholders in the global South, thus reinforcing structural inequalities. In the case of the Changing Worlds study, the initial research idea did emerge from UK-based researchers, and we were aware of the potential for actual or perceived inequity when establishing international partnerships. However, there were a number of safeguards to promote just and equitable collaboration. The practices we adopted largely map onto the key domains outlined by Faure et al. [23]. Several aspects addressed in previous sections include control over funding and information export, support for capacity building with bidirectional knowledge sharing, and data ownership. Additional factors that support equity include fair attribution of intellectual contributions, as expected under the principles of research integrity. Authorship of resulting manuscripts was never assumed a priori and determined only on the basis of actual contributions to each specific research output. The local teams made decisions on authorship and were prioritised in the author order. In hindsight, there were also domains where improvements could be made. As mentioned, we did not create formal research agreements. Explicit documentation of respective responsibilities can be a critical tool in securing fair processes and outcomes for all involved. Instead of formal arrangements, collaborations were and continue to be based on trust and respect at the interpersonal level, which is also a crucial building block to supportive and effective research partnerships. However, constructive professional relationships between individuals or small teams may still not mitigate the asymmetrical power dynamics that are rooted in institutional distrust (e.g., prejudiced attitudes towards the capacity and skills of organisations in the Global South). To ensure that the mutual commitments and contributions towards the research partnerships were recognised, we featured the collaboration on Climate Cares' social media channels and the institutional website. We also made efforts to disseminate the research findings in joint presentations, most notably as a side event at the Planetary Health Alliance meeting in Boston, 2022. This allowed us to articulate and celebrate each team's achievements, as well as the collaborative successes. The various teams continue to collaborate on an ad hoc basis as opportunities arise and to meet resourcing needs to complete scheduled outputs, while some continuing collaborations have also emerged that are connected to a coherent (albeit not formally articulated) research programme. Establishing more formal agreements may become pertinent if shared competitive funding is obtained to support the next stages of the Changing Worlds research program.

Finally, we acknowledge that there is tension between the goal of delivering culturally appropriate and contextually relevant survey research and the goal, for analytical reasons, of achieving comparability in the quantitative metrics. While we maintained significant overlap between the survey instruments in the Changing Worlds study, there were some additions, omissions, and variations in the measures included by the different site teams. Thus, when interpreting between-site differences on key outcome variables, we must take into account these site-specific methodological variations. It may not always be possible to firmly distinguish between a true effect of sociocultural context and a spurious effect due to differences in survey delivery, which is a limitation that must be acknowledged. Moreover, some of our primary measures, such as the Climate Distress Scale and the Climate Agency Scale, still require further psychometric validation, including cross-cultural testing. The uncertain reliability and validity of these metrics across time and space adds an element of noise and thus warrants caution in interpreting similarities and differences in our findings across the study sites. Some of the newer climate anxiety measures (e.g., the Climate Change Anxiety Scale [24] and the Hogg Eco-anxiety Scale [25]) have since been used in different countries [14,26–31]. These measures show promise in terms of capturing and comparing climate change-related anxiety across samples, although further clarification of their psychometric properties is still required [32]. Our findings suggest that different scales (e.g., climate distress scale, eco-anxiety scale) capture related but nevertheless distinct experiences, including cognitive, behavioural, physical, and emotional aspects. More work is needed not only to understand how the various metrics relate to each other but, more

fundamentally, to clarify the conceptual underpinnings of these constructs [33,34]. Much of the existing conceptual and psychometric validation work remains restricted to developed countries. There is a need for global collaboration among methodological and subject matter experts in order to determine the appropriateness and comparability of existing measures and—possibly—to develop novel measures that can provide adequate and meaningful cross-cultural data on climate-related psychological responses. These endeavours are costly and time-consuming, requiring substantive international funding. Given the urgency and potentially escalating nature of the problem, we need more sustained funding to not simply develop metrics to capture the impacts but also to use those outputs to inform and evaluate (mental health) interventions.

*4.2. Lessons Learned*

One evident lesson that emerged from each of the study sites was the importance of working with local young people to deliver research outcomes. When the Changing Worlds project originated in 2020, evidence of the psychological impact of climate change had started to emerge only anecdotally. While practitioners and researchers expressed particular concern for the mental health of young people, a strong empirical basis was still lacking. We also noted that despite the growing movement of young climate activists, youth tend to remain excluded from decision-making on climate change and are overlooked as agents of change [35]. We strongly believe it is important to recognise that young people are the experts in their lived experience, and that they should—at the very least—be included or consulted in the design stages of research into climate anxiety. We were able to recruit exceptionally motivated, diverse groups of young people to provide input on the survey design and delivery at each site. This step was invaluable in contextualising each instance of the survey and ensuring that locally meaningful, actionable insights can be derived. We also recognise that there are intersecting layers of vulnerability that affect young people in different parts of the world and in different socioeconomic settings. Because of the direct input from young people, we were able to capture not only the extent of climate-related worry and anxiety but also how this was experienced in relation to timely and contextually relevant stressors (e.g., COVID-19, economic recession, political instability) that may have a multiplicative effect on youth wellbeing. In the future, if the project is sustainably resourced, we would also consider creating opportunities to meet and share learnings between the different youth advisory groups to support cross-cultural exchanges and capacity building around the social justice aspects of climate change.

The Changing Worlds studies were designed to address specific research questions around the occurrence of climate-related psychological responses and the relationship with mental health, but also to allow the exploration of emerging patterns of emotions, impacts, and behaviours. Thus, in addition to generating comparative data, the site teams were also able to identify specific local knowledge gaps, research opportunities, and resource requirements. The UK survey, being the first in the series, showed that climate distress was more pronounced than pandemic-related distress, which was remarkable at the height of the pandemic and in a setting where few people had direct experience with climate impacts. Climate change seems to loom large in the minds of young people, even in the face of other global crises. The USA data highlighted how direct experience of climate-related hazards (or the perception thereof) can heighten many aspects of young people's psychological response to climate change. While our study was the largest in the USA to date, the observed behavioural pattern requires further validation in a nationally representative sample in order to gauge the true prevalence of these more intense responses and whether certain climate events are more likely to engender perspective changes among youth. Among Caribbean youth, generalised anxiety levels were fairly high and life satisfaction was rated as moderate. Distress around the pandemic and climate change appeared to contribute to anxiety symptoms. In the context of projected climate impacts in the region, urgent research is needed in order to understand the intersecting vulnerabilities that may result in poor wellbeing outcomes under increased climate-related pressures. In India,

the investigations revealed a need for more awareness-raising on climate change and its impacts, as this was notably limited among young people living in slums, especially in places that were not directly affected by climate change. The team's observations also indicated that young people needed more opportunities and platforms from which to take action in order to allow them to tap into their sense of agency to support climate adaptation efforts and to protect their communities. In the Philippines, the study development stage and consultation with youth representatives made it clear that stress and anxiety about the outcome of the 2022 national election could potentially compound anxiety around climate change and worry about their futures. The upcoming survey provides a unique opportunity to generate insights into potentially intersecting sources of mental health and wellbeing impacts among Filipino youth, centred around feelings of grief and injustice which are shared aspects of climate, pandemic, and political stress. Our efforts notwithstanding, much more research is needed in order to understand the nature and scope of the effect of climate change on youth globally, and efforts to amplify the voices of those most vulnerable to climate threat are crucial. Ongoing research should focus on the meaningful and active involvement of young people in order to ensure that the insights we gather are actionable and support positive wellbeing outcomes and involvement in climate action.

*4.3. Envisioning a New Path Forward*

While this study succeeded in bringing together expertise across regions to deliver novel insights into the ways in which youth are navigating their "changing worlds", this is only a small step towards understanding how youth mental health is affected by climate change and what can or should be undertaken to support resilience among this population. The emerging responsive space exploring climate change and mental health remains siloed, disconnected, and unjust, as health and climate inequities go unaddressed. Recognising this, the Wellcome Trust has recently funded an innovative project that aims to catalyse research and action at this exact intersection (https://www.connectingclimateminds.org (accessed on 17 July 2023)). Ambitiously scoped, the project will cultivate a more connected, supported, and engaged community of practice on a global scale in order to create a research and action agenda. Young people are one of the identified vulnerable groups and key stakeholders that will be actively engaged in the regional and global dialogues that will set the research agenda. The authors of this paper are all involved in various roles with this important work, which extends and scales up the efforts to create an evidence base that incorporates lived experiences and that truly recognises and responds to community needs. The effective working relationships forged through the "Changing Worlds" study have been a crucial component in progressing the Connecting Climate Minds initiative, highlighting the value of building international collaborative relationships, and they will play an important role in seeing it through to completion.

In addition, the team behind the Changing Worlds study also co-designed a guided journal for young people using participatory design principles. The resulting journal was implemented in a physical journal booklet with daily reflection and writing activities that encourage the diarist to explore their thoughts and feelings around climate change, to identify self-care routines, and to find specific actions they can take. With a dearth of validated interventions for people experiencing climate distress, the UK team conducted a small pilot study to obtain initial feedback from a target population and ascertain the feasibility of larger-scale intervention studies with this journal. A similar pilot study is currently underway in the Caribbean, led by the same team that delivered the Changing Worlds study in this region and supported by the UK team. Additional intervention studies are planned in Australia and the Philippines in 2024.

These examples highlight how understanding and responding to the mental health and wellbeing implications of the climate crisis on global youth is best achieved not just through collaborations between research teams, but with meaningful public engagement. Wherever possible, we should centre lived experience and implement participatory methodologies. Achieving these goals will require substantive resources to be mobilised, including further

investment from committed funders who recognise the importance of relationship and trust building as crucial steps in achieving actionable outcomes that truly meet the needs of the communities affected by climate change.

## 5. Conclusions

The Changing Worlds study aimed to explore psychological responses to climate change, the experience of climate distress, the mental health impacts of climate change, and the perception of agency among youth in parts of the world that have been differentially affected by climate change. The work was sensitive to the premise that different sociocultural norms might affect how concern about climate change is perceived and expressed, and explored the distinct combinations of economic and political pressures that were at play during a global pandemic. This research collaboration between relatively independently functioning site teams struck a balance between producing comparative data and allowing for site-specific amendments that met local needs, interests, and logistical constraints. The products of this research collaboration have highlighted how distress around climate change is common among youth, with impacts on their decision making capacity and mental health, and also how local challenges shape young people's psychological responses to the climate crisis. This paper summarised the teams' experiences and the lessons we learned in the process. We call for the research community to continue engaging in interdisciplinary and international research that also actively involves young people; their voices are critical in producing actionable evidence for incorporating health and wellbeing outcomes as essential components of global, regional, and local climate policies.

**Author Contributions:** A.V. wrote the first draft of this manuscript and created the graphs and tables; S.K.Y. and M.D. created the data summaries for the India study; S.M., M.H.C. and N.G. created data summaries for the Caribbean study; R.G., J.J.B.A. and C.A.P. created the data summaries for the Philippines study; B.W., A.V. and E.L.L. created the data summaries for the USA study; E.L.L. project managed the Changing World Study. All authors have read and agreed to the published version of the manuscript.

**Funding:** UK team: The study was supported by the Institute for Global Health Innovation. E.L.L. was supported by the Lenore England Innovation Fund at the Institute of Global Health Innovation. USA team: B.W. was supported by postdoctoral funding from the Stanford Center for Innovation in Global Health and the Stanford Woods Institute for the Environment, Stanford University. Additional project funding was provided by the Billion Minds Institute. Caribbean team: The study was funded by The University of the West Indies Grant Number Campus Research and Publication Fund—CRP.3.MAR21.08. India team: S.K.Y. is supported by the Adolescents' Resilience and Treatment Needs for Mental health in Indian Slums (ARTEMIS) project funded by the UK Research and Innovation/Medical Research Council (UKRI/MRC), Grant no: MR/S023224/1, and M.D. is supported partly under the Systematic Medical Appraisal, Referral, and Treatment for Common Mental Disorders in India—SMART Mental Health project funded by NHMRC (Grant No: APP1143911) and partly by the International Study of Discrimination and Stigma Outcomes (Indigo) funded by MRC, Grant no: MR/R023697/1.

**Data Availability Statement:** Data from the UK survey are freely available online (https://osf.io/mgu6x). Survey data from the other study sites may be requested from the authors. At the time of writing no data had been collected for the Philippines study site.

**Acknowledgments:** We would like to thank the members of the various YPAG's for their invaluable contributions to the survey development. We also acknowledge the skilled contributions of the field staff in India who conducted the interviews. Quinta Seon, who was a Queen Elizabeth Fellow from McGill University at UWI, provided support with data analysis for the Caribbean study.

**Conflicts of Interest:** The authors have no conflict of interest to declare.

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
