# Peer review of "Investigating the Mental Health Impacts of Climate Change in Youth: Design and Implementation of the International Changing Worlds Study"

_challenges, doi:10.3390/challe14030034_

Round 1

Reviewer 1 Report

Thank you for the opportunity to read this manuscript of important current relevance. The authors have presented the project in question in detail and comprehensively with a lot of information and data. However, I think it could be given a more practical slant that would help researchers and institutions. In this regard, I would remove the section on funding as I feel it is too long and does not make the manuscript easy to read. Moreover, they seem to me to be less important aspects for scientific relevance. It would also be important to create a section to indicate possible practical interventions through which to engage young people about the climate change challenge, with a focus on stakeholder and community involvement through participatory methodologies between institutions, academics and the community. 

Author Response

Reviewer 1:

Thank you for the opportunity to read this manuscript of important current relevance. The authors have presented the project in question in detail and comprehensively with a lot of information and data.

We thank the reviewer for their support and for positive assessment of our work’s current relevance.

However, I think it could be given a more practical slant that would help researchers and institutions. In this regard, I would remove the section on funding as I feel it is too long and does not make the manuscript easy to read. Moreover, they seem to me to be less important aspects for scientific relevance.

While we agree that details on the funding arrangements are not of specific scientific relevance, we do think that they are of practical/operational significance. Since this is a project report outlining the design and implementation of this project, we do think it is valid to include some reference to this. However, we accept that this should not be the emphasis of the paper, so we have cut back the text in that section (see the new abbreviated Section 2.2.2.), keeping the focus of this section on project management rather than the financial arrangements that supported the project, which are outlined in the required funding statements as part of the acknowledgements (see Lines 823-835).

It would also be important to create a section to indicate possible practical interventions through which to engage young people about the climate change challenge, with a focus on stakeholder and community involvement through participatory methodologies between institutions, academics and the community. 

We agree with this, and we have now highlighted in Section 4.3. how the research partnerships we formed through the Changing Worlds study are supporting additional research, including intervention studies based on participatory action principles (Lines 772-790).

Reviewer 2 Report

 Both the title and the abstract and the keywords reflect the contents of the text. References up to the year 2023 are reflected in the bibliography.

The article submitted for evaluation is important for the scientific community, since it explains the effects of climate change on people's mental health. However, the text is a narration of the characteristics and phases of the investigation, the results themselves are not collected, only the detailed explanation of the project itself.

Author Response

Reviewer 2:

Both the title and the abstract and the keywords reflect the contents of the text. References up to the year 2023 are reflected in the bibliography.The article submitted for evaluation is important for the scientific community, since it explains the effects of climate change on people's mental health. However, the text is a narration of the characteristics and phases of the investigation, the results themselves are not collected, only the detailed explanation of the project itself.

We appreciate the reviewer’s assessment of the importance of the topic we address with this paper. As the reviewer correctly points out, what we present here is intended as a “project report”, focused on analysing our methodology rather than the empirical findings. The primary goal of this manuscript was to describe the design and implementation of this international study, highlighting challenges and lessons learned along the way. Our main contribution with this paper is that our experiences and reflections may be of use to others looking to engage in similar collaborations. In a future publication we intend to elaborate on the empirical observations.

We have edited the title and made some changes to the abstract to make the primary goal of this manuscript clearer, from the outset.

Reviewer 3 Report

This is an impressive study and provides a great foundation for future work. But have the authors made the most of this opportunity to compare and discuss the emerging findings? This is said to be beyond the scope of the present manuscript (l 402) - but it is not explained why:

1. I realise that individual papers are already being published from different locations describing results but I feel that the authors have missed an opportunity to provide more of an overview of results as well as the design principles and methodologies. In particular fig 1 is rather rudimentary and it may be better to replace with a Table providing more detail on what was actually found. And the discussion of emerging findings (l 522-546) is also rather brief. I suggest that the authors consider discussing more detail e.g. any variation in effects according to age, gender, socio-cultural factors etc. 

2. Even where specific results are noted, there is little discussion of interpretation or significance. For example, the UK (fig 1 and l 527-) reports that climate distress exceeds pandemic stress whereas the impact on daily functioning was moderate for the pandemic yet mild for climate change. Why this disconnect? Is it a feature of the perceived timescale of effect? More generally, why do high levels of concern on climate change not translate into greater degree of individual action - could this be related to the emotion of despair?

3. The authors emphasise that better understanding of the comparative mental health impacts is needed in order to develop future research questions (section 4.3) to inform policy and practice. While the future expansion of the work in a Wellcome Trust-funded study sounds exciting, it is still the case that further discussion of the results here, provisional though they may be, would help other researchers, including in other localities, who may not be part of that study.

4. It would also help to see a little more discussion of whether results from the different surveys could be influenced by variation in survey content/emphasis on different emotion words e.g. between UK-India-US. This is discussed in relatively theoretical terms (section 4.1) but concrete examples would help.

I have a couple of other specific comments:

5. As the Philippines survey is not yet completed and no outputs included, what is the rationale for including this country here?

6. Formation of YPAGs is a great part of the design. Was each YPAG selected from a larger number of respondents to the invitation? If so, what were the criteria for selection and were they similar in the different geographical initiatives?

7. Section 2.3.3 raises the very important issue of potential impediment of EU GDPR on international data sharing for research - this concern needs to be made more visible to regulators, researchers and their institutions, and research funders. While the authors might feel that this policy issue is tangential to the main conclusions from their manuscript, the GDPR impediment will continue problematic for new, larger international studies and I suggest that this concern is emphasised more strongly. 

Author Response

Reviewer 3:

This is an impressive study and provides a great foundation for future work. But have the authors made the most of this opportunity to compare and discuss the emerging findings? This is said to be beyond the scope of the present manuscript (l 402) - but it is not explained why:

We thank the reviewer for their positive assessment of this work. We understand the point regarding the relative lack of detail on the emerging findings, and on the comparison between study sites. As the reviewer has correctly summarised, we do not wish to duplicate the work that is led and published by the individual teams, based in the countries where the surveys have taken place. The site teams have priority in terms of publishing their own findings, which is also indicated in the outputs section of Table 1.

Our aim with the current paper was to focus on the process of design and implementation, rather than the empirical data and project outcomes. This approach also reflects the objective of the side event that we hosted at the PHA meeting in 2022, on which this paper is based. To make this clear from the start, we have edited the title and we have attempted to more explicitly state our intentions in the revised abstract.

  1. I realise that individual papers are already being published from different locations describing results but I feel that the authors have missed an opportunity to provide more of an overview of results as well as the design principles and methodologies. In particular fig 1 is rather rudimentary and it may be better to replace with a Table providing more detail on what was actually found. And the discussion of emerging findings (l 522-546) is also rather brief. I suggest that the authors consider discussing more detail e.g. any variation in effects according to age, gender, socio-cultural factors etc. 

We agree that a more thorough investigation of the similarities and differences between the sites will be very insightful and perhaps direct future research. Further assessment of the impact of demographic and socio-cultural variables on the measures outcomes is also warranted. We are planning to write a separate paper that focuses on this when the Philippines team has concluded their data collection. At this point in time, we wanted to focus on the procedural element of this collaboration and reflect on the implications of our methodology. The planned empirical paper and this current paper are designed may be read as companion pieces. This two-stage approach is allowing us to delve deeper into both the methodological aspects and the resulting data. Combining this into single paper would not allow us to fully explore both aspects in the level of detail that we would like to share with the readers.

  1. Even where specific results are noted, there is little discussion of interpretation or significance. For example, the UK (fig 1 and l 527-) reports that climate distress exceeds pandemic stress whereas the impact on daily functioning was moderate for the pandemic yet mild for climate change. Why this disconnect? Is it a feature of the perceived timescale of effect? More generally, why do high levels of concern on climate change not translate into greater degree of individual action - could this be related to the emotion of despair?

While we stand by our approach (as outlined in response to the previous comment), we agree that the reader might benefit from gaining some insight into the data that is being generated and the themes we are will be looking to explore in further detail once we have all the empirical data (including data from the survey planned in the Philippines). We now elaborate on empirical observation highlights from Figure 1 (see lines 494-562). This helps to illustrate how the methodology allowed us to uncover context-specific findings around shared themes, bringing into focus how climate distress is experienced and how it might impact day-to-day wellbeing and mental health in slightly different ways, depending on the situation. However, we stress that this is not based on modelling or quantitative comparisons, but rather on qualitative evaluations of the different teams’ key findings. As mentioned earlier, we plan to formally and quantitatively compare the findings between the sites.

  1. The authors emphasise that better understanding of the comparative mental health impacts is needed in order to develop future research questions (section 4.3) to inform policy and practice. While the future expansion of the work in a Wellcome Trust-funded study sounds exciting, it is still the case that further discussion of the results here, provisional though they may be, would help other researchers, including in other localities, who may not be part of that study.

We appreciate this point. As mentioned in response to the previous comment, we have expanded our discussion of the empirical results to provide richer context for the reader to understand the kinds of themes that we will look to explore in future work based on the provisional findings and observations of the site teams.

  1. It would also help to see a little more discussion of whether results from the different surveys could be influenced by variation in survey content/emphasis on different emotion words e.g. between UK-India-US. This is discussed in relatively theoretical terms (section 4.1) but concrete examples would help.

We agree that this is a potential contributor to differential outcomes, however without formal analysis we think it is inappropriate to comment on this in a manner that goes beyond hypothetical/theoretical. We now explicitly comment on the fact that the results presented here reflect a qualitative examination rather than formal statistical comparisons (Lines 499-502) and on the need to further examine differences in the affective characterisation of climate change impacts (Lines 550-552).

I have a couple of other specific comments:

  1. As the Philippines survey is not yet completed and no outputs included, what is the rationale for including this country here?

As stated earlier, our primary goal was to report on the process of designing and delivering this project, not on the outcomes (i.e., the empirical results). While the implementation of the Philippines study was delayed relative to the other sites, the site team was crucially involved in the Changing Worlds study. We think it is important to note their contributions to the design process even if the outcomes are not yet available for this site. We now also acknowledge the site/study location in Figure 1, to ensure that we are consistently including this site as relevant to the current paper despite the fact that we have yet to commence data collection there. We have also provided additional detail on that study site in Table 1.

  1. Formation of YPAGs is a great part of the design. Was each YPAG selected from a larger number of respondents to the invitation? If so, what were the criteria for selection and were they similar in the different geographical initiatives?

Yes, we can confirm that at most sites, YPAG participants were selected from a larger pool of respondents either to an open Expressions of Interest call, or through more targeted advertisement of the opportunity to relevant youth organisations such as student groups and youth climate or social justice advocacy groups. However, we did not apply formal selection criteria across the board. The site teams generally aimed to achieve a reasonable gender balance within each YPAG, and sought to include participants from different cultural/ethnicity and socio-economic groups relevant to the country, and where possible also a fair geographic spread, including people from rural and urban areas. We also asked prospective YPAG members to indicate their availability and motivations for participation, to identify individuals that were sufficiently engaged to ensure a productive YPAG process. In India we made use of an existing adolescent expert advisory group (AEAG) that was set up for another cRCT (ARTEMIS). We have provided additional details in the revised manuscript (Lines 342-352).

  1. Section 2.3.3 raises the very important issue of potential impediment of EU GDPR on international data sharing for research - this concern needs to be made more visible to regulators, researchers and their institutions, and research funders. While the authors might feel that this policy issue is tangential to the main conclusions from their manuscript, the GDPR impediment will continue problematic for new, larger international studies and I suggest that this concern is emphasised more strongly. 

We understand that data sharing limitations across international borders is a concern for potential collaborators. Our recommendations are that the project plan includes time and resources to consult with legal professionals on this matter in the early stages or project development. Most institutions will have this internal capacity, but the process of creating compliant data sharing agreements between institutions can be time consuming. We have rephrased this point to emphasise the importance of planning this aspect of the project (Lines 439-463).

Reviewer 4 Report

Thank you for putting this report together on the Changing Worlds Study and for describing in great detail, and with transparency, the process, challenges, and results of the work. It was very informative and accessible and provided a template of research design for others to consider and build upon. This seems to be much needed research, and I'm sure your continued contributions to the field will be invaluable.Noted a typo on line 472. Sentence appears to be missing an 'and'.

Author Response

Reviewer 4:

Thank you for putting this report together on the Changing Worlds Study and for describing in great detail, and with transparency, the process, challenges, and results of the work. It was very informative and accessible and provided a template of research design for others to consider and build upon. This seems to be much needed research, and I'm sure your continued contributions to the field will be invaluable. Noted a typo on line 472. Sentence appears to be missing an 'and'.

Thank you for your supportive comments. Unfortunately, we could not locate the typo on Line 472.